# Impact on beer sales of removing the pint serving size: An A-B-A reversal trial in pubs, bars, and restaurants in England

**Eleni Mantzari**[1,2], **Gareth J. Hollands**[3], **Martin Law**[4,5], **Dominique-Laurent Couturier**[4,5], **Theresa M. Marteau**[1] *

1 Behaviour and Health Research Unit, University of Cambridge, Cambridge, United Kingdom, 2 Department of Health Services Research and Management, City, University of London, London, United Kingdom, 3 EPPI Centre, UCL Social Research Institute, University College London, London, United Kingdom, 4 MRC Biostatistics Unit, University of Cambridge, Cambridge, United Kingdom, 5 Papworth Trials Unit Collaboration, Royal Papworth Hospital NHS Foundation Trust, Cambridge, United Kingdom

* tm388@cam.ac.uk

**Data Availability Statement:** All data are available from the Open Science Framework here: https://osf.io/afweh.

**Funding:** The work of this report was funded in whole by Wellcome [PI: TMM: ref 206853/Z/17/Z (Collaborative Award in Science: Behaviour Change

## Abstract

### Background

Smaller serving sizes could contribute towards reducing alcohol consumption across populations and thereby decrease the risk of 7 cancers and other diseases. To our knowledge, the current study is the first to assess the impact on beer, lager, and cider sales (hereafter, for ease, referred to just as "beer sales") of removing the largest draught serving size (1 imperial pint) from the options available in licensed premises under real-word conditions.

### Methods and findings

The study was conducted between February and May 2023, in 13 licensed premises in England. It used an A-B-A reversal design, set over 3 consecutive 4-weekly periods with "A" representing the nonintervention periods during which standard serving sizes were served, and "B" representing the intervention period when the largest serving size of draught beer (1 imperial pint (568 ml)) was removed from existing ranges so that the largest size available was two-thirds of a pint. Where two-third pints were not served, the intervention included introducing this serving size in conjunction with removing the pint serving size. The primary outcome was the mean daily volume of all beer sold, including draught, bottles, and cans (in ml), extracted from electronic sales data. Secondary outcomes were mean daily volume of wine sold (ml) and daily revenue (£). Thirteen premises completed the study, 12 of which did so per protocol and were included in the primary analysis. After adjusting for prespecified covariates, the intervention resulted in a mean daily change of −2,769 ml (95% CI [−4,188, −1,578] p < 0.001) or −9.7% (95% CI [−13.5%, −6.1%] in beer sold. The daily volume of wine sold increased during the intervention period by 232 ml (95% CI [13, 487], p = 0.035) or 7.2% (95% CI [0.4%, 14.5%]). Daily revenues decreased by 5.0% (95% CI [9.6%, −0.3%], p = 0.038).

by Design: Generating and Implementing Evidence to Improve Health for All)]. The funders had no role in the study design, data collection and analysis, decision to publish, or preparation of the manuscript.

**Competing interests:** The authors have declared that no competing interests exist.

## Conclusions

Removing the largest serving size (the imperial pint) for draught beer reduced the volume of beer sold. Given the potential of this intervention to reduce alcohol consumption, it merits consideration in alcohol control policies.

## Trial registration

ISRCTN.com ISRCTN18365249.

---

Author summary

### Why was the study done?

- Removing the largest serving size of wine by the glass (usually 250 ml) reduces the volume of wine sold in licensed premises.

- It is unknown whether removing the largest serving size of other alcohol drinks, such as beer, has a similar effect.

### What did the researchers do and find?

- We asked 13 licensed premises in England to remove the offer of their largest serving size of draught beer (1 imperial pint, 568 ml) from available options for 4 weeks. We compared the total volume of beer sold during the intervention period to that sold during the nonintervention periods.

- Removing the largest serving size for draught beer (the imperial pint) reduced the daily mean volume of beer sold by 9.7%.

### What do these findings mean?

- This intervention merits consideration for inclusion in alcohol control policies.

- The findings are limited due to a lack of assessment of all alcoholic drinks sold in participating licensed premises, which would have allowed for consideration of whether people may have fully compensated for their reduced beer consumption by drinking other alcoholic drinks.

## Background

Alcohol consumption contributes to premature mortality and preventable morbidity, causing millions of deaths annually worldwide and ranking fifth among 20 risk factors for the global

burden of disease [1,2]. Reducing alcohol consumption across populations is therefore a global public health priority [3].

Aspects of physical and economic environments influence alcohol consumption across populations. These include the marketing [4–7], labelling [8–10], availability [11–15], and affordability of alcohol products [16,17]. Most implemented interventions to date have sought to reduce the affordability of alcohol and to control their marketing and licensing [18,19]. Another promising intervention includes reducing the size of servings and packages of products that can harm heath, including alcoholic drinks. People consume less food and nonalcoholic drinks when presented with smaller portions, packages, or tableware [20–22], while the package and container size of alcoholic drinks can influence alcohol consumption. Smaller wine glasses in restaurants decrease the volume of wine sold and accordingly consumed [23]. Smaller wine glasses might also reduce the amount of wine drunk at home [24]. Drinking wine at home from smaller bottles, compared with standard 75 cl bottles, may also reduce consumption when the bottles are 50 cl [25], but the impact of 37.5 cl bottles is less certain [24].

Interventions that target the sizes of servings to reduce alcohol consumption fall into 2 groups [26]:

1. removing the largest serving size(s) from existing options;

2. adding a smaller size to existing options, which could be larger or smaller than the existing smallest serving size (s).

Either of these interventions could be implemented alone or in combination. For example, if targeting draught beer in licensed premised in the United Kingdom, implementing the first would involve removing the pint (568 ml), without altering any other serving size options. Alternatively, this intervention could be combined with the second option, so that a smaller serving size not normally offered, such as two-thirds of a pint, is added to the range. Similarly, this second option could be implemented alone, for example, by adding the two-third pint serving size to the existing range of options without removing the larger serving size.

The limited evidence to date suggests that the second option—adding a smaller size to existing options—does not impact alcohol consumption. This is supported by findings from a recent study, conducted in real-world settings, in which adding a serving size of draught beer of two-thirds of a pint that was between the largest (1 imperial pint) and the smallest size (half a pint) across 13 licensed premises had no discernible effect on the volume of beer sold [27]. In contrast, removing the largest serving size from existing options (option i) alone or in combination with the second option (adding a smaller serving size) has shown potential for reducing alcohol consumption.

Two studies, one conducted in a laboratory and one in a seminaturalistic context of a pub, found reductions in alcohol consumed on a single occasion when larger servings were removed and replaced by smaller sizes [28]. In the only study assessing this in a real-world setting in licensed premises, removing the largest serving size of wine by the glass (most often 250 ml) for 4 weeks decreased wine sales—a proxy for consumption—by 7.6% [29]. It is unknown whether removing the largest serving size of other alcohol drinks, such as beers, would have similar effects. We are also unaware of any studies that have assessed the impact of adding a size smaller than the existing smallest serving size on alcohol consumption.

In the UK, the largest and most popular serving size of draught beer is the imperial pint [30]. At 568 ml, this size is larger than the typical sizes for draught beer found in most other countries. For example, in the United States of America, the largest and most popular size is 473 ml [31]. In the Netherlands and Belgium, the usual serving size is 250 ml; in France, it is 330 ml; and in Germany, it is 500 ml, depending on the region and type of beer ordered [30].

In many parts of Australia, the most common serving size is the "schooner," i.e., 425 ml. In South Australia, this size is called "a pint" and is therefore the biggest most popular size on offer [32]. As suggested by news reports, licensed premises in Western Australia have undergone a process by which a pint-drinking culture has been replaced by a schooner-drinking one, to tackle increasing costs and adhere to government efforts to reduce alcohol consumption. We are unaware of any relevant scientific research that corroborates this [33–35].

The aim of the current study is to assess the impact on the volume of beer, lager, and cider sold of removing the largest serving size of draught beer, lager, and cider (1 imperial pint, 568 ml) so that the largest serving size available is two-thirds of a pint. We hypothesise that this intervention reduces the volume of beer, lager, and cider sold.

## Methods

The study was approved by the University of Cambridge Psychology Research Ethics Committee (reference no: PRE.2022.103). The study protocol was preregistered (ISRCTN: ISRCTN18365249 (https://www.isrctn.com/ISRCTN18365249); Open Science Framework: registration: https://osf.io/369sh/, protocol https://osf.io/shckx, statistical analysis plan https://osf.io/n7pdy).

### Study design

The study used an A-B-A treatment reversal design consisting of 3 consecutive 4-week periods. "A" represented the nonintervention periods during which usual serving sizes were offered, and "B" represented the intervention period during which the largest serving size (pint) of draught beer, cider, and lager—hereinafter referred to collectively as "beer" for ease—was removed from premises' existing range so that the largest serving size available was two-thirds of a pint.

### Setting and context

The study was conducted in pubs, bars, and restaurants in England, where draught beer must legally be available in at least 1 of 2 sizes [36]: pint (568 ml), which is the most popular measure [30], and half-pint (284 ml). Since 2011, one-third (189 ml) and two-third pint servings (379 ml) can also be sold, but there is no legal obligation for these to be available [30,37].

### Participating premises

Thirteen licensed premises in England participated in the study. Their characteristics are shown in Table 1. The majority of these were in London (61.5%), located in more deprived areas, with 77% falling within the first and second Index of Multiple Deprivation quintiles.
Eligibility criteria for premises' participation were the following:

1. sell a minimum of 150 pints of beer on average per week;

2. be willing to remove the larger serving of draught beer, i.e., a pint, and introduce a two-third pint if this serving size was not already available;

3. have an electronic point of sale (EPOS) till system to record daily sales of all drinks and their served sizes;

4. be primarily indoor, permanent establishments in a fixed location, i.e., not purposefully temporary or time-limited (e.g., pop-up) or mobile venues (e.g., vans).

**Table 1. Characteristics of participating licensed premises.**

| Premises number | Location | Index of Multiple Deprivation Quintile* | Premises type** | Baseline daily revenue (£; mean (SD)) | Offered 2/3 pint before study |
|---|---|---|---|---|---|
| 1 | Hackney, London | 1 | Restaurant | 5,831.6 (2,870.9) | No |
| 2 | Hackney, London | 1 | Pub | 1,101.9 (938.5) | No |
| 3 | Haringey, London | 5 | Pub | 1,149.6 (687.5) | No |
| 4 | Wandsworth, London | 1 | Cocktail bar and Restaurant | 8,223.6 (7,968.4) | Yes |
| 5 | Brighton and Hove | 1 | Pub | 1,530.7 (1,038.4) | No |
| 6 | Newham, London | 3 | Bar and Restaurant | 967.2 (441.0) | Yes |
| 7 | Birmingham | 2 | Restaurant | 3,681.7 (3,042.1) | Yes |
| 8 | Lewisham, London | 2 | Bar and Restaurant | 2,351.2 (1,072.3) | No |
| 9 | Lewisham, London | 3 | Bar | 993.9 (798.4) | No |
| 10 | Hackney, London | 2 | Pub | 1,676.0 (1,831.0) | No |
| 11 | Brighton and Hove | 1 | Restaurant | 1,510.5 (990.7) | No |
| 12 | Sheffield | 1 | Bar | 675.1 (256.3) | Yes |
| 13 | Brighton and Hove | 2 | Bar | 1,177.3 (676.8) | No |

*The Index of Multiple Deprivation (IMD) ranks every area in England according to deprivation levels. The IMD combines information from the 7 domains to produce an overall relative measure of deprivation; 1 = most deprived; 5 = least deprived.

**Description of premises type taken from each premises' website.

## Sample size calculation

A simulation-based predictive power analysis was performed based on data from a previous study assessing the impact of removing the largest serving size of wine by the glass in 21 licensed premises, using an A-B-A design with each period lasting 4 weeks [29]. This previous study found that the intervention resulted in a mean 7.6% (95% CI [−12.3%, −2.9%]) reduction in the daily volume of wine sold. The simulations suggested that 5 licensed premises would be required for the current study to detect an effect of this size on the daily volume of beer sold with a probability of 0.85 at the 0.05 significance level. Due to the model complexity [38], it was considered favourable to increase the sample size to 10 premises. To account for possible attrition, 13 premises were recruited. The sample size calculation report providing a more thorough description of the process used to define the number of premises is available as Supporting information (S2 Appendix).

## Intervention

Licensed premises removed the largest serving size of draught beer (1 imperial pint) from their existing ranges so that the largest serving size available was two-thirds of a pint. Where two-third pints were not usually served, this serving size was introduced, with proportionate pricing as far as possible, i.e., with a price that was linear-by-volume between the pint and half-pint sizes. Premises were provided with the necessary two-third pint glassware by the research team. Menus and signs were updated to reflect the changes.

Within the TIPPME intervention typology for changing environments to change behaviour [39], the type of intervention used in the current study is classified as "Size," focused on the "Product" itself (i.e., the alcoholic drink(s), as opposed to, for example, aspects of the wider environment).

## Measures

**Primary outcome.** Daily volume (in millilitres (ml)) of all beer, lager, and cider (draught as well as bottled and canned), extracted from electronic records of sales.

**Secondary outcomes.** The following outcomes were extracted from electronic records of sales from each premises:

1. Number of beers and cider sold in each serving size per day:

   - one-third pint (189 ml) draught

   - half-pint (284 ml) draught

   - 330 ml bottle

   - 375 ml bottle

   - two-third pint (379 ml) draught

   - 440 ml can

   - 500 ml bottle

   - pint (568 ml) draught

   - 568 ml can

   - 550 ml bottle

   - 660 ml bottle

   - 750 ml bottle

2. Daily volume (in ml) of wine sold, in order to assess whether the absence of the largest serving of beer affects wine consumption, given that beers, cider, and wines make up more than 70% of alcoholic drinks sold in licensed premises in the UK [40];

3. Daily revenue from food and all drinks, alcoholic and nonalcoholic.

**Covariates.** Given that daily temperature, day of the week, season, and holidays can influence alcohol sales [41,42], the following covariates were considered:

1. Maximum daily local temperature;

2. Special events (including national events, i.e., Valentine's Day, St Patrick's Day, Mother's Day, Bank Holidays—Good Friday, Easter weekend, May Day—and local events, i.e., DJ nights, quiz nights; live music events, parties; open mic/karaoke evenings, fundraiser events, local markets, marathons and half-marathons, and beer festivals);

3. Total revenue;

4. Day of the week;

5. Time in days from start of the study.

**Procedure.** Potentially eligible licensed premises were identified through a publicly available database (www.whatpub.com). Invitations to participate in the study were sent to those based in 1 of 6 geographical areas to which the research team could readily travel in order to conduct fidelity checks. Those expressing interest were sent more information about the study

and then assessed for eligibility over the telephone. Eligible premises wishing to participate provided written informed consent to take part.

Premises changed their available serving sizes for draught beer on 2 occasions over a period of 12 weeks: once to remove pints and introduce two-third pints (if this serving size was not already available); and once to reintroduce pints and remove two-third pint serving sizes (if this serving size had been added during the intervention period). Till systems were updated to reflect the new serving sizes.

Premises were contacted 1 day before each reversal to remind them of the required changes. Fidelity to the protocol was checked by visits organised by the research team in the first week after each occasion on which changes were due to occur. No premises failed any fidelity checks, but given these were conducted only once following each reversal, sales data were also checked for any protocol violations.

Data were collected between February 2023 and May 2023. Premises were paid £3,000 (including VAT) to compensate for expected losses to revenue as well as the resources needed for them to take part in the study, including the timely provision of all requested data. Premises were additionally reimbursed for any costs associated with changes to menus and signs.

## Data analysis

For the primary analysis, a heteroscedastic linear mixed model was used to predict the cube root of daily volume of beer sold as a function of the following fixed effects: (i) study period (reference nonintervention period); (ii) day of week (using contrasts of type sum); (iii) study day (per-site standardised time from start of the study); (iv) total daily revenue (per-site standardised (log) revenue); (v) standardised maximum daily temperature; and (vi) special events (dichotomous variable with 0 for normal days (reference) and to 1 for special events). The analysis excluded any days when premises were closed.

Possibly correlated random intercept and standardised (log) revenue slope for each site were used to take the within site-dependence into account. The cube root was used to linearise the relationship between response and predictors, as well as to overcome heteroscedasticity. All model checks suggested a good fit of the assumed model to the data. The residual variance was allowed to differ for site and day of week. Estimates were obtained by maximising the restricted maximum likelihood, via the function lme of the nlme package of the R Statistical Software (version 4.4.0) [43].

Data included in the final analysis were from those sites that completed the study in full per protocol.

## Sensitivity analyses

To check the robustness of the primary conclusions from models when aggregating the 3 sets of 4-week conditions, 4 sets of sensitivity analyses were conducted:

**Sensitivity analysis 1.** Modelling the primary analysis, using a heteroscedastic linear mixed model, with study period as fixed-effects predictor without controlling for other predictors while modelling the dependence (random effects) and heteroscedasticity as in the primary analysis.

**Sensitivity analysis 2.** Same model as in the primary analysis, using a heteroscedastic linear mixed model, but applied to all available daily-level data, i.e., including data from sites with incomplete data and/or whose data indicated a violation of protocol for intervention implementation, e.g., sales of pints during period B.

**Sensitivity analysis 3.** Same model as in the primary analysis, using a heteroscedastic linear mixed model, but considering the intervention predictor as a 3-level factor (with levels A1, B, and A2), thus allowing the pre- and postintervention periods to have different average sale levels.

**Sensitivity analysis 4.** Use of a paired *t* test to compare the premises' beer sales aggregated per intervention period, obtained by considering the mean daily sales. Like sensitivity analysis 1, this model does not control for other predictors. Furthermore, the model relies on fewer assumptions than the primary analysis, suggesting that the conclusions of the primary analysis are not dependent on its assumptions.

## Analysis of secondary outcome

Three sets of secondary analyses were conducted:

1. Negative binomial regression analysis to estimate the number of beer drinks sold in each serving size per day according to study Period (A versus B).

2. A heteroscedastic linear mixed model analysis, similar to that used for the primary analysis (i.e., using the fixed and random effects) to estimate the daily volume of table wine (excluding fortified wines) sold according to study period (A versus B).

3. A heteroscedastic linear mixed model analysis, similar to that used for the primary analysis (i.e., using the fixed and random effects) to estimate total daily revenue according to study Period (A versus B).

## Results

The flow of premises through the study is shown in Fig 1. Thirteen licensed premises were recruited from 1,740 contacted in 6 geographical areas of England, a recruitment rate of 0.75%. Site 2 violated the protocol by selling pints during the intervention period, identified by inspection of their data. All data from this premises were excluded from the primary analysis.

### Primary outcome analysis: Volume of beer sales

The unadjusted mean daily volume of beer sold per 12 premises during the nonintervention periods (A1 + A2) was 43,136 ml (SD = 48,592.4) and 36,061.0 ml (SD = 43,099.8) during the intervention period (B) (Table 2). After accounting for prespecified covariates (day of the week; study day; total revenue; temperature; special events), there was a significant effect of study period (Table 3): The mean daily change in volume of beer sold was −2,769.21 ml (95% CI [−4,188.46, −1,577.75], $p < 0.001$) or 9.7% (95% CI [−13.5%, −6.1%]) during the intervention period (B) compared to the 2 nonintervention periods (A1 + A2). Fig 2 shows the effect of the intervention on beer sales overall and for each of the 12 premises included in the primary analysis. Detailed time series plots, showing daily beer, wine, and total sales by site, are shown in the Supporting information (S1 Appendix, Figs A1 to A13).

### Sensitivity analyses

Results and conclusions were unchanged when running the model with our 2-level intervention factor as a unique predictor. Indeed, the simplified model led to an average change in daily beer sales of −1,535.3 ml (95% CI [−2,342.6, −863.4], $p < 0.001$) or −13.8% (95% CI

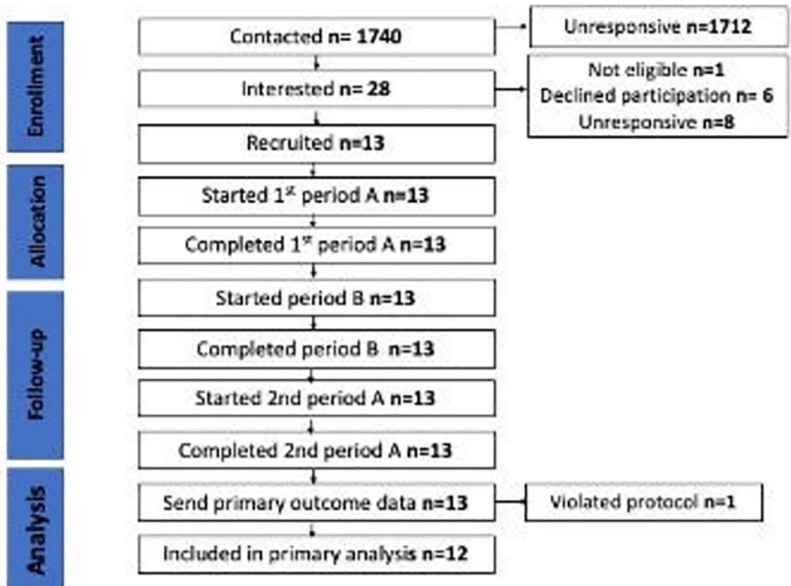

**Fig 1. Flow of premises through the study.**

[−19.0%, −8.8%]) during the intervention period (B) compared to the nonintervention periods (A1 + A2) (Table B in S3 Appendix).

Similarly, an intention-to-treat analysis ($n$ = 13) that included the one premises that had violated the protocol had no effect on the main results. The model fit showed that, on average, −2,912.7 ml (95% CI [−4,386.0, −1,685.3], $p$ < 0.001) or −9.6% (95% CI [−13.2%, −6.1%]) was the change in beer sold per day during the intervention period (B) compared to the non-intervention periods (A1 + A2) (Table C in S3 Appendix).

**Table 2. Unadjusted mean (SD) volume (ml) of beer sold per day, overall and by serving size, and volume of wine (ml) sold of premises included in primary analysis ($n$ = 12).**

|  | Nonintervention periods [A1 + A2] (both combined) | Intervention period [B] |
|---|---|---|
| **Overall volume of beer sold** | 43,476.8 (48,592.4) | 36,061.0 (43,099.8) |
| **Volume of beer sold in 1/3 pints (189 ml)** | 88.3 (404.6) | 78.0 (331.2) |
| **Volume of beer sold in 275 ml** | 0.99 (16.5) | 2.93 (28.3) |
| **Volume of beer sold in 1/2 pints (284 ml)** | 2,127.4 (3,020.0) | 2,218.4 (2,889.5) |
| **Volume of beer sold in 330 ml** | 3,212.1 (19,239.8) | 2,085.7 (7,731.3) |
| **Volume of beer sold in 355 ml** | 0 | 2.53 (42.3) |
| **Volume of beer sold in 375 ml** | 21.0 (131.1) | 10.7 (83.2) |
| **Volume of beer sold in 2/3 pints (378 ml)** | 666.3 (2,294.9) | 28,714.5 (37,571.3) |
| **Volume of beer sold in 440 ml** | 1,203.6 (4,651.9) | 1,320.0 (5,208.4) |
| **Volume of beer sold in 500 ml** | 1,384.0 (2,633.0) | 1,485.8 (3,191.7) |
| **Volume of beer sold in 550 ml** | 17.9 (143.3) | 15.6 (173.2) |
| **Volume of beer sold in 568 ml cans** | 6.17 (83.6) | 40.4 (216.2) |
| **Volume of beer sold in pints (568 ml)** | 34,713 (41,338.6) | 0 |
| **Volume of beer sold in 6,600 ml** | 2.39 (39.7) | 0 |
| **Volume of beer sold in 750 ml** | 29.9 (219.3) | 21.3 (124.9) |
| **Volume of wine sold** | 4,706.7 (5,255.3) | 4,822.1 (4,611.6) |

**Table 3. Heteroscedastic linear mixed model main results, estimating the volume (ml) of beer sold per day on the transformed (cube root) scale, on the original scale and on the relative scale, covariate coefficients shown in Table A in S3 Appendix (*n* = 12).** Rounding is to 2 decimal places.

| Scale | Intercept | Intervention | | | |
|---|---|---|---|---|---|
| | Estimate | Estimate | Lower 95% CI | Upper 95% CI | *p*-value |
| Transformed (cube root) | 30.62 | −1.01 | −1.39 | −0.64 | |
| Original (ml) | 29,126.66 | −2,769.21 | −4,188.46 | −1,577.75 | <0.001 |
| Relative (%) | 100.00 | −9.66 | −13.52 | −6.08 | |

In order to assess whether the 2 nonintervention periods were comparable, an additional analysis was conducted in which the 2 nonintervention periods were added to the model separately. The results showed that sales of beer did not significantly differ during the 2 nonintervention periods (A1 versus A2) (*p* = 0.374) (Table D in S3 Appendix), justifying the choice of combining data from both nonintervention periods for the primary analysis.

The analysis considering per intervention period aggregated sales data per site (obtained by estimating the average of the daily sales per site and intervention period) rather than daily-level data also concluded that beer sales were significantly lower during the intervention period (B) compared to the nonintervention periods (A1 + A2) (average change of −7,862 ml or −20%; *p* < 0.001) (Table E in S3 Appendix).

## Analysis of secondary outcomes

**Beer sales by serving size.** During the nonintervention periods, the largest selling serving size in terms of volume (Table 2) and average number of drinks sold per day (Table F in S3 Appendix) was the pint (568 ml). During the intervention period, the largest selling serving size was the two-third pint (Table 2 and Table F in S3 Appendix).

During the intervention period, sales of half-pints, 440 ml and 568 ml cans increased slightly with statistical significance (Table F in S3 Appendix).

**Volume of wine sold.** The unadjusted mean daily volume of wine sold per premises during the nonintervention periods (A1 + A2) was 4,706.7 ml (SD = 5,255.3) and 4,822.1 ml (SD = 4,611.6) during the intervention period (B) (Table 2). After accounting for prespecified covariates (day of the week; study day; total revenue; temperature; special events), there was a

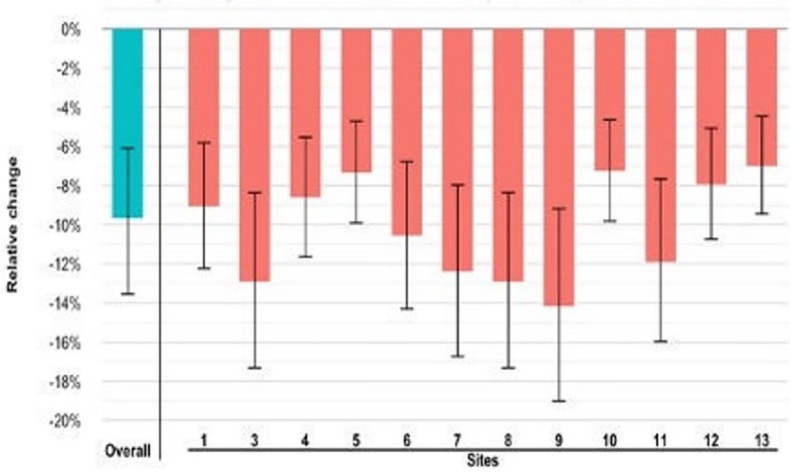

**Fig 2. Change in daily volume of beer sold (% (95% CI) with intervention.**

significant effect of study period (Table G in S3 Appendix and Fig B in S1 Appendix): On average, 232.0 ml (95% CI [112.7, 487.5], $p$ = 0.035) or 7.2% (95% CI [0.4%, 14.5%]) more wine was sold per day during the intervention period (B) compared to the 2 nonintervention periods (A1 + A2).

**Daily revenue.** The unadjusted mean daily per premises sales during the nonintervention periods (A1 + A2) was £2,336.9 (SD = 3,053.6), and £2,091.5 (SD = 2,703.8) during the intervention period (B). After accounting for prespecified covariates (day of the week; study day; total revenue; temperature; special events), there was a significant effect of study period (Table H in S3 Appendix): On average, daily revenues decreased during the intervention period (B) compared to the nonintervention periods (A1 + A2) (−£67.2; 95% CI [−£146.6, −£3.7], $p$ = 0.038); −5.0% (95% CI [−9.6%, −0.3%]).

## Discussion

Removing the largest serving size of draught beer (the imperial pint, 568 ml) from the range of options available in 12 licensed premises reduced the volume of beer sold by 9.6%. The intervention was also associated with a small absolute increase in the volume of wine sold during the intervention period, and a small decrease in daily revenues.

The intervention had the hypothesised effect of reducing the volume of beer sold. This is in keeping with recent findings showing that removing the largest serving size of wine in licensed premises reduced the volume sold by a similar amount: 7.6% (95% CI [−12.3%, −2.9%]) [29]. It is also consistent with the results of 2 studies conducted in seminaturalitic contexts—which found a reduction in alcohol consumption on a single occasion when larger servings were removed and replaced by smaller sizes [28]—and with a large body of evidence on the effect of smaller serving sizes on food consumption [44].

The results suggest that when the largest serving size of draught beer was not available, people shifted to the next available size, the two-third pint, which resulted in them drinking less. This could be explained by a tendency to consume a specific number of "units" (e.g., number of glasses or bottles), regardless of serving or package size [45]. If patrons ordered a preset number of beer servings, regardless of size, with less alcohol per serving in two-third pints, they would purchase less overall. There were also more half-pints sold during the intervention period, suggesting that some patrons might have shifted to this size in the absence of pints, but this did not appear to have happened enough to compensate for the removal of the 1 pint serving size.

Two-thirds of a pint is arguably not too small a measure to be considered a large deviation from 1 pint and thus provoke resistance, but small enough to reduce consumption. Had the largest size available been the half-pint instead, that may have been considered too small, in part because there appears to be a negative attitude in England, albeit declining, towards ordering and drinking half-pints [46]. This could therefore inadvertently have led to greater consumption than that observed in the current study [47,48]. Another reason people tend not to prefer smaller servings or packages is that these are not proportionately priced, so provide less value for money [25]. Premises in the current study were asked to price two-third pints in proportion to pint and half-pint sizes to ensure they represented the same value for money compared to other available draught serving sizes. While all premises confirmed they had done so, we were only able to verify this for the small number of premises that had drinks lists available.

In addition to the intervention having the predicted impact on lowering beer sales, it was associated with a small increase in the absolute volume of wine sales. We considered 3 possible explanations for this unexpected finding, informed by reflections from some of the participating premises. First, it is a chance finding, perhaps due to factors unrelated to the study. Second,

it is a result of some beer drinkers switching to wine when pint servings are not available. This seems more plausible in premises that serve food and for people who tend to have beer before, and wine with, their meal. In the absence of their typical serving size for beer, such customers might have opted immediately for wine. This was the account given by one of the premises, which was responsible for around 50% of the increase in wine sales observed during the study. Interestingly, however, following the reintroduction of pints, wine sales in this premises were still rising, casting doubts on this explanation. Third, it is a result of premises managers responding to reduced beer sales by promoting wine sales. Two premises reported changing their wine lists during the intervention period and promoting their new lists at that time.

However, even if the increase in wine sales in the absence of the largest serving size for beer is not a chance finding, it is important to note that this effect on wine was small in absolute terms and far smaller than the effect of reduced sales of beer. Removing the largest serving size resulted in an average per premises per day of 2,769.2 ml less beer being sold, equivalent to approximately 5 pints or 13 units of alcohol, if assuming an average ABV for beer of 4.6% [49]. The volume of wine sold increased 232.0 ml per day, i.e., almost 1 large glass of wine or approximately 3 units of alcohol—assuming an average ABV for wine of 12% [49]. After accounting for the small increase in wine sales, the intervention therefore still resulted in approximately 10 fewer units of alcohol being sold per day by each premises. Given that no level of alcohol consumption is currently considered safe for health [50], such a reduction could meaningfully contribute to population health.

In terms of the study strengths, to our knowledge, this is the first study to estimate the impact on sales of removing the largest serving of size of draught beer in pubs, bars, and restaurants, under real-world conditions. The pint has been the standard and most popular serving size for beer in England for centuries [30], having been established in 1698 [51]. Indeed, "going for a pint" has become synonymous with the act of going for a drink in British culture [52]. Previous attempts to remove this "iconic" serving size for research purposes have been unsuccessful [29]. Further strengths of this study include the use of objective measures to assess the primary and secondary outcomes, i.e., electronic records of sales and the high retention rate of premises through the study.

The study, however, also has several limitations. First, it was unable to fully assess the possibility of any compensation effects. Due to the complexity of the sales reports provided by participating premises, it was not feasible to assess sales of all alcoholic drinks. Although beers, ciders, and wines are estimated to contribute more than 70% of the sales of alcoholic drinks in licensed premises in the UK [40], it was not possible to assess sales of spirits or cocktails, estimated to contribute to the remaining 30%. It is not known, therefore, whether people might have compensated for their reduced beer consumption by drinking more of these other alcoholic drinks. Given that the intervention also resulted in a decrease in daily revenue and that cocktails and spirits tend to be more expensive than beers, this seems unlikely. It is also not known whether customers compensated for the removal of pints by drinking stronger beers, i.e., those higher in percentage alcohol by volume (% ABV). We judged this unlikely given there were very small variations in the % ABV of beers sold by premises, with most options ranging from 4% to 5%. Also, beers lists did not change during the study, and none of the premises managers interviewed at the end of the study reported any changes in the drinking patterns of their customers during the intervention period relating to beer strength. Finally, it is unknown whether people compensated for their reduced beer consumption by drinking more alcohol at home, although this has not been shown to be the case in previous research, in which, like in the current study, pints were replaced by two-third pints during one drinking occasion in a seminaturalistic setting [28]. Second, caution is needed in generalising these findings. The low response rate to an invitation to participate in the study might have resulted

in sampling bias, meaning that participating premises may not have been representative of typical licensed premises in England. Additionally, the majority of premises were in London, which potentially restricts the generalisability of the findings to this large metropolitan city. The remainder were also in cities and no premises were located in smaller towns or rural areas. Third, sales were used as a proxy for actual consumption; direct measurement of which at scale in these kinds of real-world settings is not feasible. Sales are, however, a valid proxy for consumption [53] and are commonly used in behavioural research [54–56]. Finally, the intervention was assessed for a 4-week period, leaving uncertainties about whether the observed effects are sustained over time.

In terms of the broader implications of this study, the sizes of servings of alcoholic drinks sold in licensed premises in England are subject to regulations, and draught beer must be legally available for sale in pints and half-pints [36]. One-third and two-thirds of a pint can also be sold, but licensed premises are not legally obliged to offer these [30,37]. The pint is by far the most popular serving size in the UK [30]. Based on the current findings, removing this serving size from the range offered in licensed premises and replacing it with two-thirds of a pint could contribute to policies for reducing alcohol consumption at the population level, thereby meriting consideration as part of alcohol control policies. Given that alcohol contributes between 5% and 10% of energy intake among those who consume it [57,58], the intervention also merits consideration as part of policies tackling obesity in adults.

The results suggest a possible unexpected small increase in wine sales associated with the intervention, an effect that requires replication. Given evidence that removing the largest serving size of wine by the glass from the range offered in licensed premises reduces wine sales without affecting beer sales [29], regulations might be considered that target the largest servings of both beer as well as wine. Indeed, the impact of simultaneously removing the largest serving sizes of both draught beer and wine by the glass should be assessed, whether as part of monitoring the impact of a change in regulations or in further field studies.

Interventions that involve removing or reducing serving or package sizes are generally less supported by the public than information-based interventions such as health warning labels [59–61]. Given that the pint has been the customary serving size for draught beer in England for centuries [30], significant pushback for its removal was expected both by the research team and premises managers. But premises reported receiving surprisingly few comments or complaints from customers when the largest serving size was reduced to two-thirds of a pint. Four of the 13 participating premises reported receiving some complaints, which abated as customers got used to the new serving sizes. Whether this was because customers knew that the change was time-limited or because they realised that two-thirds of a pint was sufficient remains to be explored. Indeed, regulating serving sizes in licensed premises could help shift social norms for what constitutes an appropriate serving size [62–64], both for consumption out of the home such as in pubs and bars, as well as for consumption at home where most drinking occurs [65]. This possible indirect effect of the intervention awaits study. Given that serving sizes perceived as normal do not lead to compensation effects, in contrast to when sizes are perceived as small [66], future research should also assess the possible mediating effect of social norms about serving sizes on the potential for people to compensate for reduced consumption in licensed premises by drinking more at home; although previous has shown this not to be the case.

Although customers did not voice strong objections to the removal of pints in the 13 premises that participated in this study, the low response rate to invitations to participate in the study—fewer than 1% of premises approached, lower than the 1.2% rate observed in a previous similar study for wine [29]—suggests that support for the intervention might be generally low. Expected loss of revenue increases the possibility that licensed premises would object to

intervention implementation should it be considered as part of alcohol regulation policies. Although removal of pints in the present study reduced daily revenues, losses were relatively small (average change of −£67 (5% of total revenues), 95% CI [−£146.6, −£3.7]). After the study ended, none of the participating premises removed their pint servings. Without regulation, this intervention is most unlikely to be implemented. Regulation to implement it will—understandably—meet resistance from the alcohol industry, a resistance seen to minimum unit price policies [67], but a resistance that needs to be addressed for effective alcohol control policies.

In conclusion, removing the largest serving (the imperial pint) for draught beer from the range of options available in licensed premises, so that the largest size became two-thirds of a pint, reduced the volume of beer sold. Given the potential of this intervention to reduce alcohol consumption, it merits consideration in alcohol control policies.

## Supporting information

**S1 Appendix. Addtional figures.**
(DOCX)

**S2 Appendix. Sample size calculation report.**
(PDF)

**S3 Appendix. Addtional figures and analyses.**
(DOCX)

**S1 CONSORT Checklist. Checklist of information to include when reporting a randomised trial.**
(DOC)

## Acknowledgments

We would like to thank the premises taking part in this study for their excellent cooperation. We would also like to thank Ms. Caveny Mantzaris for her help in cleaning and checking the data as part of her interniship at the Behaviour and Health Research Unit, University of Cambirdge in May/June 2023.

## Transparency declaration

The lead author (the manuscript's guarantor) affirms that the manuscript is an honest, accurate, and transparent account of the study being reported; that no important aspects of the study have been omitted; and that any discrepancies from the study as originally planned (and, if relevant, registered) have been explained.

## Author Contributions

**Conceptualization:** Eleni Mantzari, Gareth J. Hollands, Theresa M. Marteau.

**Formal analysis:** Martin Law, Dominique-Laurent Couturier.

**Investigation:** Eleni Mantzari.

**Methodology:** Eleni Mantzari, Gareth J. Hollands, Theresa M. Marteau.

**Supervision:** Eleni Mantzari.

**Validation:** Eleni Mantzari.

**Writing – original draft:** Eleni Mantzari.

**Writing – review & editing:** Eleni Mantzari, Gareth J. Hollands, Theresa M. Marteau.

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
