## [Editor Report · Decision Letter 0]

4 Mar 2024

Dear Dr Marteau, 

Thank you for submitting your manuscript entitled "Impact on beer sales of removing the pint serving size: an A-B-A reversal trial in pubs, bars and restaurants in England" for consideration by PLOS Medicine.

Your manuscript has now been evaluated by the PLOS Medicine editorial staff as well as by an academic editor with relevant expertise and I am writing to let you know that we would like to send your submission out for external peer review.

However, before we can send your manuscript to reviewers, we need you to complete your submission by providing the metadata that is required for full assessment. We also have the following editorial request: please include the study protocol document as approved by your committee as Supporting Information. 

To this end, please login to Editorial Manager where you will find the paper in the 'Submissions Needing Revisions' folder on your homepage. Please click 'Revise Submission' from the Action Links and complete all additional questions in the submission questionnaire.

Please re-submit your manuscript within two working days, i.e. by Mar 07 2024 11:59PM.

Feel free to email me at kjanin@plos.org or contact the journal office at plosmedicine@plos.org if you have any queries relating to your submission.

Kind regards,

Katrien G. Janin, PhD

Senior Editor

PLOS Medicine

---

## [Decision Letter · Decision Letter 1]

10 Apr 2024

Dear Dr. Marteau,

Thank you very much for submitting your manuscript "Impact on beer sales of removing the pint serving size: an A-B-A reversal trial in pubs, bars and restaurants in England" (PMEDICINE-D-24-00685R1) for consideration at PLOS Medicine. 

Your paper was evaluated by a senior editor and discussed among all the editors here. It was also sent to independent reviewers, including a statistical reviewer. The reviews are appended at the bottom of this email and any accompanying reviewer attachments can be seen via the link below:

[LINK]

After discussing the paper with the editorial team, I’m pleased to invite you to revise the paper in response to the reviewers’ comments. We plan to send the revised paper to the original reviewers, and we cannot provide any guarantees at this stage regarding publication. 

When you upload your revision, please include a point-by-point response that addresses all of the reviewer and editorial points, indicating the changes made in the manuscript and either an excerpt of the revised text or the location (eg: page and line number) where each change can be found. Please also be sure to check the general editorial comments at the end of this letter and include these in your point-by-point response. When you resubmit your paper, please include a clean version of the paper as the main article file and a version with changes marked as a marked-up manuscript. Please also check the guidelines for revised papers at http://journals.plos.org/plosmedicine/s/revising-your-manuscript for any that apply to your paper

We expect to receive your revised manuscript by May 01 2024 11:59PM. However, if this deadline is not feasible, please contact me by email, and we can discuss a suitable alternative. 

Please use the following link to submit the revised manuscript: https://www.editorialmanager.com/pmedicine

Don’t hesitate to contact me directly with any questions (kjanin@plos.org). If you reply directly to this message, please be sure to ‘Reply All’ so your message comes directly to my inbox. 

We look forward to receiving your revised manuscript. 

Sincerely,

Katrien Janin, PhD

PLOS Medicine

plosmedicine.org

kjanin@plos.org

Please include line numbers in your revised manuscript, ideally not starting from 1 with each new page.

Please cite the reference numbers in square brackets. Citations should precede punctuation.

Abstract: Please report your abstract according to CONSORT for abstracts, following the PLOS Medicine abstract structure (Background, Methods and Findings, Conclusions) https://www.equator-network.org/reporting-guidelines/consort-abstracts/ (see per previous study “Impact on wine sales of removing the largest serving size by the glass: An A-B-A reversal trial in 21 pubs, bars, and restaurants in England’ - https://doi.org/10.1371/journal.pmed.1004313)

At this stage, we ask that you include a short, non-technical Author Summary of your research to make findings accessible to a wide audience that includes both scientists and non-scientists. The Author Summary should immediately follow the Abstract in your revised manuscript. This text is subject to editorial change and should be distinct from the scientific abstract. Ideally each sub-heading should contain 2-3 single sentence, concise bullet points containing the most salient points from your study. In the final bullet point of ‘What Do These Findings Mean?’, please include the main limitations of the study in non-technical language. Please see our author guidelines for more information: https://journals.plos.org/plosmedicine/s/revising-your-manuscript#loc-author-summary. 

Please complete the CONSORT checklist and ensure that all components of CONSORT are present in the manuscript. When completing the checklist, please use section and paragraph numbers, rather than page numbers.

Discussion: please present and organize the Discussion as follows: a short, clear summary of the article's findings; what the study adds to existing research and where and why the results may differ from previous research; strengths and limitations of the study; implications and next steps for research, clinical practice and/or public policy implications; followed by a one-paragraph conclusion. Please remove all subheadings within your Discussion.

Financial Disclosure: The funding statement should include: specific grant numbers, initials of authors who received each award, URLs to sponsors’ websites. Also, please state whether any sponsors or funders (other than the named authors) played any role in study design, data collection and analysis, the decision to publish, or preparation of the manuscript. If they had no role in the research, include this sentence: “The funders had no role in study design, data collection and analysis, decision to publish, or preparation of the manuscript.”

Supplementary materials: Please note that supplementary materials are not checked and will be posted as supplied by the authors. Therefore, please double check. Please cite your Supporting Information as outlined here: https://journals.plos.org/plosmedicine/s/supporting-information - Please note you may use almost any description as the item name of your supporting information as long as it contains an "S" and number. For example, “S1 Appendix” and “S2 Appendix,” “S1 Table” and “S2 Table. Please ensure each supplementary material has a call out (link) from your main manuscript. 

To help us extend the reach of your research, please provide any Twitter handle(s) that would be appropriate to tag, including your own, your coauthors’, your institution, funder, or lab.

Comments from the reviewers:

Reviewer #1: Thanks for the opportunity to read your manuscript. My role is statistical reviewer, so I have focused on the design, data, and analysis that are presented. I have put general comments first, followed by questions relevant to a specific section of the manuscript (with a page/paragraph reference). 

This study examines the effectiveness of removing 'pint' (568 mL) serving sizes from licenced premises, replacing with a smaller size serving (2/3rds of a pint). Thirteen sites were recruited for the study. The volume of beer and cider (draught and bottled) was the primary outcome, with secondary outcomes including size-specific volume of beer/cider, daily volume of wine, and daily revenue. Data was collected over 12 week, with a 4 week baseline period, 4 weeks of intervention (pints not available), and 4 weeks of withdrawal of intervention. Data was collected through the same time-period across all sites, and fidelity to the intervention was checked at all sites. 

The main analysis used an extension of liner mixed models to allow for site-specific heterogeneity in variance. The main outcome was transformed (^1/3), with parameters in the model for study period, day of week, day of study, daily revenue, temperature, and special events. Adding daily revenue as a covariate means the main outcome for the study from this analysis can be thought of as beer/cider volume proportional to overall sales. Random intercepts were included for sites, and a random slope term for overall revenue of each site. 

Several sensitivity analyses were considered, looking at the effect of removing covariates from the main analysis, data from sites that were not adherent, allowing baseline and reversal periods to have different levels, and an analysis that directly compared total volume of beer/cider aggregated to either no-intervention or intervention. There was a reduction in the volume of beer and cider sold during the intervention period, and with a 5% decline in daily revenues and a small increase in wine sold. The treatment effect from these sensitivity analyses was close to that seen in the main analysis. 

There was a low response rate - does this affect the generalisability of the findings? 

P9. Is 'wine' just table wine or does it include fortified wines?

P12, Paragraph 1. In the methods it is mentioned that all sites passed the fidelity checks - this seems to contradict this paragraph.

P13, Table 3. I would consider some adjustments to this table. Most of the information presented here about covariates could be safely moved to an appendix. I would also consider presenting just the estimate, p-value, and 95% CI here. The intercept is also superfluous to the main research questions. Is it also possible to present a back-transformed treatment effect estimate (with CL) in both the table and the main results? This would be ideal for all results where the outcome was transformed before the analysis. 

 Supplmentary appendices. The titles/captions for the tables should have information about the type of sensitivity analysis in each case rather than just 1, 2, 3 etc.

The study has a pre-registered protocol and statistical analysis plan, these match the current manuscript.

P6, Paragraph 2. This is a comment and requires no response - I am absolutely delighted to see a mention of the strange names for beer glass sizes from South Australia. They are a complete aberration. As a frequent visitor to Adelaide I have been caught out many times by ordering a 'Schooner' (425mL in my hometown, 3/4 of a pint) and ending up with a disappointing sized glass of beer (285mL, 1/2 pint). 

P10, Paragraph 5. Just to check - does 'heteroscedastic LMM" mean that a different constraint was allowed for each site in residual variance? i.e. is this similar to what is done with the 'group' option in proc mixed in SAS or with the galamm package in R? 

P10, Paragraph 6. What checks of model fit were made?

What software was used for the analysis? 

P10, Paragraph 7. I follow the rationale for most of these sensitivity analyses, except for #4. What was assumption being tested with aggregating the data and using a paired t-test? 

P11, Paragraph 5. Did the negative binomial model just include a parameter for day of study, or did it include similar parameters to the main analysis? Was a negbin model used for every possible serving size? 

Figure 2. Instead of the 'dynamite' plot presented here, I'd consider plotting as much of the data as possible, e.g. is it possible to construct a panel graph showing the time series of primary outcome across all the study in each site? I think with the type of model used, it should also be possible to estimate a 'counterfactual' estimate of beer/cider volume sold if the intervention was not implemented (similar to an interrupted time series). 

Reviewer #2: The authors have presented an intervention study with a clear rationale robust methodology. I applaud the authors use of transparent research methods, having registered their work with the open science framework prior to publication. The authors are following up on their previous work, examining whether reducing the servicing size of wine reduced consumption but now looking at beer products. They find a similar result, lowering serving sizes does result in decreased consumption. The article is of high quality and answers an important question - while the similarities to their previous work are undeniable this is a strength of the article rather than a limitation - they are building on their prior work in a meaningful way. There are some points I would like to see the authors address ahead of publication:

1. First sentence appears to have a citation error (1), (2). Should be (1,2).

2. "Licensed premises in Western Australia have undergone a process by which a pint-drinking culture has been replaced by a schooner-drinking one, to tackle increasing costs and adhere to government efforts to reduce alcohol consumption (32-34)." - the references used here are two media publications and a Wikipedia article, the authors should use higher quality empirical evidence when discussing drinking trends and their causes. This point does not add anything of value to the manuscript and could be cut. 

3. "A simulation-based predictive power analysis" - could the authors go into more detail here, what was the exact method used, the statistical software, packages/commands. The reader should be able to replicate this process from the information provided in the text. 

4. "Thirteen licensed premises were recruited from 1740 contacted in six geographical areas of England, a recruitment rate of 0.75%." - this is incredibly low, what were the reasons that the other 1727 venues were not recruited - was this part of the recruitment strategy (e.g. contacting all venues and then recruiting the first ones that came back) or was this due to 1727 venues saying no? 

5. "ABV for beer of 4.6%" - this brings up an important point, could people have ordered a stronger beer to compensate for the smaller size? In Australia, low mid and full-strength beers are available for purchase, are these available in the UK? Could this intervention move someone from a mid to full strength beer? 

6. "The pint is by far the most popular serving size in the UK (or England?)" - the authors should choose whether it is the UK or England they are referring to. 

Reviewer #3: This is a great, well-written paper on an important topic. I enjoyed reading it. I do have some comments which I think will improve this manuscript.

Abstract

I don't think this is the first study to assess the impact of reducing serving size of beer on sales in a licensed premises. I think that is also examined in this published paper, though admittedly it is observed sales: https://onlinelibrary.wiley.com/doi/10.1111/add.14228. Yours is the first study that I am aware of to do so using objective sales data. This study is still very important and extends the prior findings considerably, but this sentence should be toned down.

Introduction

The last couple of sentences could be a little clearer, you say that a schooner is a pint, but also that Western Australia is transitioning from pints to schooners. Think this is down to regional differences in terminology but rewording could make this clearer.

It would be helpful to mention compensation in the introduction. If people have smaller drinks in the on-trade are they more likely to go home and have more drinks because they'd only had a small one? The Kersbergen study gets at this a bit. 

Theres also a theory about norms and serving size, which outlines that there is a range of serving sizes which are accepted as being one serving, https://www.ncbi.nlm.nih.gov/pmc/articles/PMC6333281/#:~:text=The%20first%20proposition%20of%20the,'not%20normal'%20in%20size whereby compensation is less likely, which I believe Dr Kersbergen has applied to alcohol consumption too. (I am not Dr Kersbergen but a big fan of her research).

What was the reasoning for only reducing one beverage type? Might it be more effective to remove all the largest servings e.g. wine, spirits.

Methods

Did you include draft cider too? You refer to draft beer throughout but seems it would make sense to include cider too? After reading the measures I think cider was included in your primary outcome, this could be clearer throughout including the title - e.g. beer and cider sales.

Why didn't you include any PPI input? I guess it's a journal requirement but feels strange to include a section if you don't actually have any PPI input, explaining why you didn't might be helpful. I also think getting some PPI input into these findings and what they actually mean in terms of practice and implementation could be a really interesting addition if you did want to incorporate some PPI.

Results

It is unclear to me why in your sensitivity analysis with intervention as a unique predictor that your mean difference is less in ml but the proportional decrease is much greater? I can't work out why the total ml would change.

Discussion

I really enjoyed your discussion, all of the things I was thinking about whilst reading the paper are unpicked and explained really nicely. (I had to go back through and delete a lot of my comments as you had done such a good job of unpicking and explaining it all!).

One additional limitation around compensation, you don't know what people drank after, did they compensate at home because they'd only had a small one?

Thanks for the opportunity to read this really interesting work - Melissa Oldham

Please upload any figures associated with your paper as individual TIF or EPS files with 300dpi resolution at resubmission; please read our figure guidelines for more information on our requirements: http://journals.plos.org/plosmedicine/s/figures. While revising your submission, please upload your figure files to the PACE digital diagnostic tool, https://pacev2.apexcovantage.com/. PACE helps ensure that figures meet PLOS requirements. To use PACE, you must first register as a user. Then, login and navigate to the UPLOAD tab, where you will find detailed instructions on how to use the tool. If you encounter any issues or have any questions when using PACE, please email us at PLOSMedicine@plos.org. 

Check FD, CI, DAS and ethics statement and include requests if necessary 

[LINK]

---

## [Decision Letter · Decision Letter 2]

5 Jul 2024

Dear Dr. Marteau,

Thank you very much for re-submitting your manuscript "Impact on beer sales of removing the pint serving size: an A-B-A reversal trial in pubs, bars and restaurants in England" (PMEDICINE-D-24-00685R2) for review by PLOS Medicine.

I have discussed the paper with my colleagues and the academic editor and it was also seen again by xxx reviewers. I am pleased to say that provided the remaining editorial and production issues are dealt with we are planning to accept the paper for publication in the journal.

[LINK]

We look forward to receiving the revised manuscript by Jul 12 2024 11:59PM.   

Sincerely,

Katrien Janin, PhD

Senior Editor 

PLOS Medicine

plosmedicine.org

Requests from Editors:

Thank you for your detailed response to the editors' and reviewers' comments. I have discussed the paper with my colleagues and the academic editor, and it has also been seen again by the original reviewers. The changes made to the paper were welcomed by the reviewers. 

I only have a few minor request for you at this stage:

1) Statistical reporting: 

We suggest reporting statistical information in the following format: ‘x’; (95% CI [‘y’,’ z’] p value). For p values, please report as p<0.001 and where higher as 'p=0.002'. Please add the statistical method used to your method section. We also invite you to report p values to consistently to the third decimal digit - thousandths. For example, see abstract : "After adjusting for pre-specified covariates, the intervention resulted in a mean daily change of -2769ml (95% CI -

4188 to -1578, p<0.00001) or -9.7% (95% CI -13.5% to -6.1%) in beer sold. The daily volume of wine sold increased during the intervention period by 232ml (95% CI 13 to 487, p=0.035) or 7.2% (95% CI 0.4% to 14.5%). Daily revenues decreased by 5.0% (95% CI -9.6% to -0.3%, p=0.038). " Please check and amend throughout. 

2) CONSORT checklist. Thank you for supplying the CONSORT checklist (S3). Please remove the page numbers from this list, and use section headers and paragraph numbers, please remove the page numbers.

3) please double check that reference #29 is updated

Comments from Reviewers:

Reviewer #1: Thanks for the revised manuscript and responses to my original review. The updated manuscript resolves my original queries - the more detailed figures in the appendix are a helpful addition

Reviewer #2: Thankyou, I am happy with the revisions and response to my comments. 

Reviewer #3: The authors have responded to all my comments and I would recommend that this paper is now published.

[LINK]

---

## [Editor Report · Decision Letter 3]

15 Jul 2024

Dear Dr Marteau, 

On behalf of my colleagues and the Academic Editor, [AE Name], I am pleased to inform you that we have agreed to publish your manuscript "Impact on beer sales of removing the pint serving size: an A-B-A reversal trial in pubs, bars and restaurants in England" (PMEDICINE-D-24-00685R3) in PLOS Medicine.

PRESS

Sincerely, 

Katrien G. Janin, PhD 

Senior Editor 

PLOS Medicine